# Aging Alters Cervical Vertebral Bone Density Distribution: A Cross-Sectional Study

Eun-Sang Moon [1], Seora Kim [1], Nathan Kim [1], Minjoung Jang [1], Toru Deguchi [1], Fengyuan Zheng [2], Damian J. Lee [2] and Do-Gyoon Kim [1,*]

[1] Division of Orthodontics, College of Dentistry, The Ohio State University, Columbus, OH 43210, USA; moon.200@buckeyemail.osu.edu (E.-S.M.); kim.4226@buckeyemail.osu.edu (S.K.); kim.6417@buckeyemail.osu.edu (N.K.); mj9@illinois.edu (M.J.); deguchi.4@osu.edu (T.D.)

[2] Division of Restorative and Prosthodontics Dentistry, College of Dentistry, The Ohio State University, Columbus, OH 43210, USA; fyzheng@gmail.com (F.Z.); lee.6221@osu.edu (D.J.L.)

* Correspondence: kim.2508@osu.edu; Tel.: +614-247-8089; Fax: +614-688-3077

**Abstract:** Osteoporosis reduces bone mineral density (BMD) with aging. The incidence of cervical vertebral injuries for the elderly has increased in the last decade. Thus, the objective of the current study was to examine whether dental cone beam computed tomography (CBCT) can identify age and sex effects on volumetric BMD and morphology of human cervical vertebrae. A total of 136 clinical CBCT images were obtained from 63 male and 73 female patients (20 to 69 years of age). Three-dimensional images of cervical vertebral bodies (C2 and C3) were digitally isolated. A gray level, which is proportional to BMD, was obtained and its distribution was analyzed in each image. Morphology, including volume, heights, widths, and concavities, was also measured. Most of the gray level parameters had significantly higher values of C2 and C3 in females than in males for all age groups ($p < 0.039$). The female 60-age group had significant lower values of Mean and Low5 of C2 and C3 than both female 40- and 50-age groups ($p < 0.03$). The reduced BMD of the female 60-age group likely resulted from postmenopausal demineralization of bone. Current findings suggest that dental CBCT can detect age-dependent changes of cervical vertebral BMD, providing baseline information to develop an alternative tool to diagnose osteoporosis.

**Keywords:** CBCT; cervical vertebra; clinical assessment; diagnosis; postmenopause

## 1. Introduction

Osteoporosis is an age-dependent systematic disease that results in bone loss, increasing the risk of fracture [1–3]. The vertebra is one of the most common anatomical sites where osteoporotic fractures were observed [4,5]. While thoracic and lumbar fractures have been frequently reported, recent clinical observations indicated that fall-induced severe cervical vertebral injuries steeply rose more than eight times for elderly population over the 47 years [6]. However, relatively limited studies have focused on osteoporosis of cervical spine.

Bone mineral density (BMD) is a primary parameter assessed to diagnose osteoporosis [7–9]. The World Health Organization (WHO) defined an osteoporotic patient as someone who has BMD with 2.5 standard deviation lower than young healthy women [10,11]. Dual energy X-ray absorptiometry (DEXA) has been used as a standard technique to diagnose BMD [7,11,12]. However, DEXA is used at low radiation doses but produces a low resolution with approximately 500 μm of two-dimensional (2D) image that only provides areal BMD [11,13] (Table 1). On the other hand, three-dimensional (3D) quantitative computed tomography (QCT) was applied to measure BMD and related parameters of cervical and lumbar vertebrae [7–9]. However, the QCT was limited due to its relatively high radiation dose and rough resolutions at the range of in-plane pixel size of 0.29 to 0.68 mm and slice thickness of 1.25 to 2.5 mm.

**Table 1.** Descriptive summary of X-ray based technologies [11].

| Technologies | Voxel Size (μm) | Effective Radiation Dose (μSv) | Scan Time (Second) |
|---|---|---|---|
| DXA | 500 | 1–20 | ~120 |
| MDCT | 156–500 | 100–8000 | <30 |
| CBCT | 130–400 | 6.3–2100 | 10–40 (Rotation) 1.92–7.2 (Exposure) |
| Micro-CT | 0.3–100 | NA | >600 |

Cone beam computed tomography (CBCT) has been widely used in a dental clinical setting [14–18]. Cervical spine, especially C2 and C3, is captured during a routine CBCT scan used in dentistry. This technique uses moderate radiation doses and provides higher resolutions (0.2 to 0.4 mm) of 3D images [17]. While it was indicated that there has been scarcity of studies that examine capability of CBCT to screen patients' BMD [19], a few studies showed that CBCT images of cervical vertebra can be used to detect osteoporosis of patients [20–22]. However, these studies are limited by using 2D region of interest that provides the relative measures of areal BMD. On the other hand, we successfully calibrated 3D CBCT images by showing the strong positive correlations between hydroxyapatite phantoms with three different densities (1000, 1250, and 1750 mg/cm$^3$) and gray levels scanned using three different resolutions (200, 300, and 400 μm) at full field of view (FOV) of CBCT (Figure 1) [11]. Thus, the objective of the current study is to examine whether dental CBCT can identify age and sex effects on volumetric BMD and morphology of human cervical vertebrae.

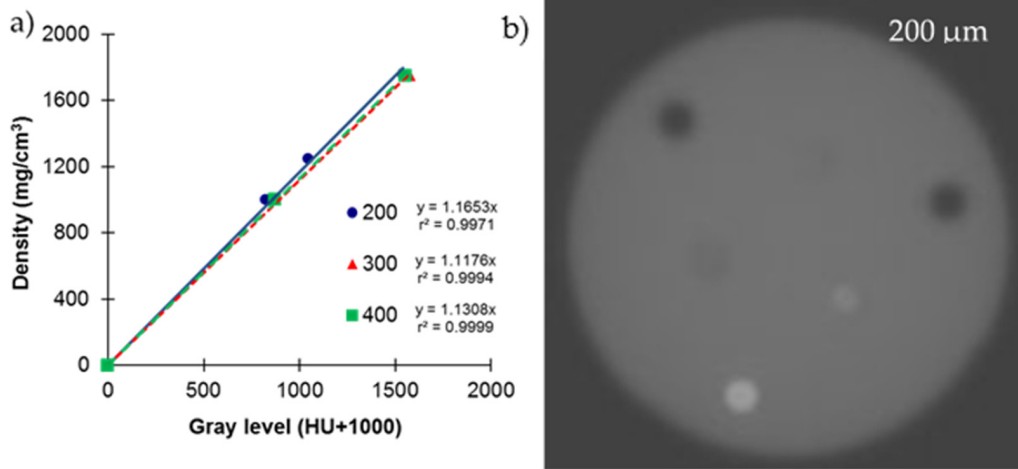

**Figure 1.** (**a**) Strong positive correlations in the calibration curves of gray values for (**b**) phantoms of bone materials (hydroxyapatite) with three different densities (1000, 1250, and 1750 mg/cm$^3$) scanned using three different resolutions (200, 300, and 400 μm) of CBCT (Reprinted with unrestricted permission for non-commercial use) [11].

## 2. Materials and Methods

The protocol used in the current study was approved by institutional review board at The Ohio State University (Protocol no. 2011H0128). A total of 136 CBCT images was randomly selected from records on routine dental patients (63 images for male and 73 images for female) at The Ohio State University, College of Dentistry. Those 3D images were taken by a CBCT machine (iCAT, Imaging Science International, Hatfield, PA, USA) with resolutions of 0.25 mm for four images, 0.3 mm for 127 images, and 0.4 mm for five images under the same scanning energy (120 kV and 5 mA) with full field of view (FOV) at different scanning times (26.9 s, 8.9 s, and 8.9 s for images with 0.25 mm, 0.3 mm, and 0.4 mm voxel sizes, respectively). These scanning conditions are routinely used in

clinical practices. The exclusion criteria for the CBCT images were craniofacial anomalies, vertebral anomalies (fusions and major vertebral asymmetries), orthognathic patients with bone plates, and limited field of view images without second and third cervical vertebrae. Three age groups including 40-age group (20 to 49 years old, 36.61 ± 11 years), 50-age group (50 to 59 years old, 54.55 ± 2.94 years), and 60-age group (older than 60 years old, 64.81 ± 2.82 years) were assigned for male (21 images for each age group) and female (29 images for 40-age group, 24 images for 50-age group, and 20 images for 60-age group). The sample size was determined using a previous clinical observation that showed the significant changes of CBCT based cervical vertebral mean gray levels with aging (1948.31 ± 81.87 vs. 1997.26 ± 50.03) [18]. Fifteen CBCT images turned out to provide the minimum number of samples needed to obtain significant results ($p < 0.05$) with 80% statistical power. Therefore, the current number of CBCT images used for each age group was assumed to be sufficient to satisfy the power of statistical analysis.

The 3D CBCT images were imported to image-analysis software (ImageJ, NIH) (Figure 2a), and two cervical vertebrae (C2 and C3) were digitally cropped and saved to individual image files. Non-bone voxels outside of vertebra were removed using semi-automatic heuristic algorithm [16,18,23]. Posterior and lateral processes were cut from 10 voxels away from the endplates of vertebral body producing an integral volume of vertebral body. A gray level of each voxel, which is proportional to BMD, was obtained (Figure 2e) and collected for its histogram of C2 and C3 (Figure 2f,g). The values of Mean, standard deviation (SD), and low and high gray levels at the 5th and the 95th percentiles (Low$_5$ and High$_5$) of voxel counts in the histogram were determined (Figure 2f).

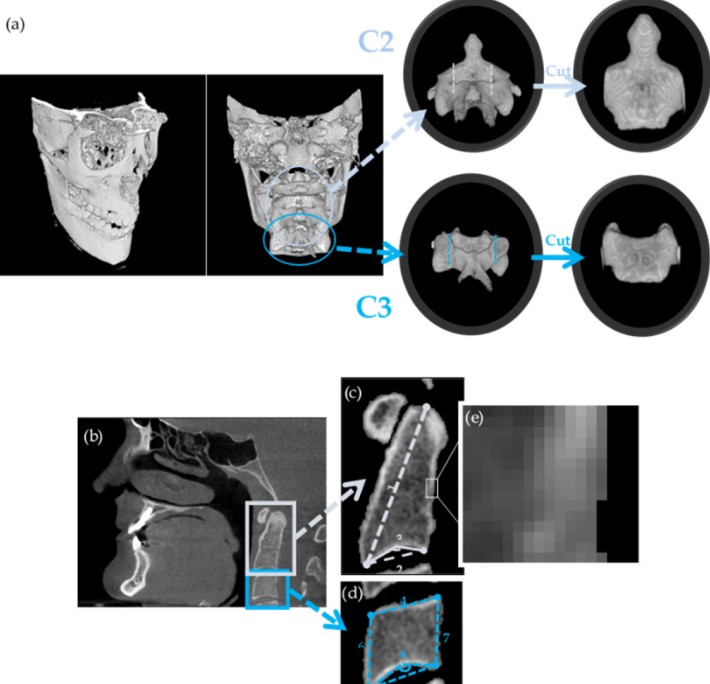

**Figure 2.** *Cont.*

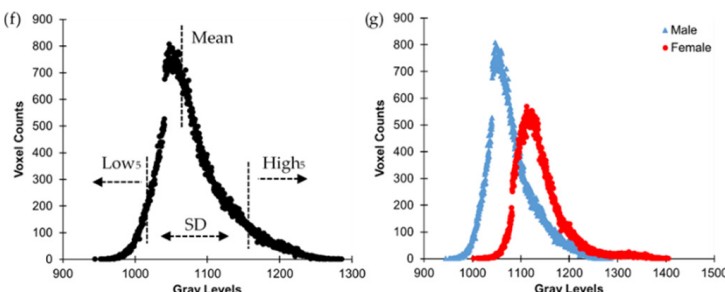

**Figure 2.** (**a**) Digital isolating process of C2 and C3 in 3D CBCT images for (**b**) the mid-sagittal CBCT slide views for (**c**) C2 and (**d**) C3 to measure (1) C2 height, (2) C2 width, (3) C2 concavity, (4) C3 top width, (5) C3 bottom width, (6) C3 anterior height, (7) C3 posterior height, and (8) C3 concavity. (**e**) Heterogeneity of gray levels, (**f**) a typical gray level histogram with gray level parameters, and (**g**) a comparison of the histograms between male and female. Male had higher voxel counts (i.e., volume) and female had higher gray levels (i.e., BMD parameters).

Morphology of C2 and C3 were measured at a single mid-sagittal CBCT slide. The CBCT images were aligned at the orientation of the cervical vertebrae using imaging software (Dataviewer, Bruker, Kontich, Belgium). The aligning process consisted of the following: (1) odontoid process of second cervical vertebrate should coincide with the mid-sagittal plane of the head, (2) transverse process of the second cervical vertebrae should be coplanar, and (3) posterior surface of the odontoid process should be aligned with the coronal plane of the head. Then, a mid-sagittal slide that bisects the odontoid process of C2 was selected (Figure 2b) and reference points on the vertebra were placed following a protocol modified from a previous study [24]. The reference points were placed at the most convex areas. The morphological parameters including C2 height, C2 width, C2 concavity, C3 top width, C3 bottom width, C3 anterior height, C3 posterior height, and C3 concavity were measured using ImageJ (Figure 2c,d).

### 3. Statistical Analyses

Reliability tests for measurements of all parameters (five samples each) were made using intra- and inter-rater agreements with intra-class correlation coefficient (ICC). Pearson correlations were tested between all parameters. Analysis of variance (ANOVA) followed by Tukey HSD (honestly significant difference) post hoc testing was also utilized to compare all of the parameters between age and sex groups for C2 and C3. The significant level is set at $p < 0.05$.

### 4. Results

The ICCs for all of BMD parameters were higher than 0.997 for both intra- and inter-rater agreements ($p < 0.001$) and those for the morphological parameters were higher than 0.763 ($p < 0.039$).

The C2 had significantly higher values of SD, volume, and height but significantly lower values of Mean, $Low_5$, and concavity than the C3 ($p < 0.014$). The width of the C2 was measured significantly wider than the top of the C3 but significantly narrower than the bottom of the C3 ($p < 0.001$). The value of $High_5$ was not significantly different between C2 and C3 ($p = 0.433$). Most of values of the gray level and morphological parameters were significantly correlated between C2 and C3 ($p < 0.001$) except concavity in male ($p = 0.116$). In particular, the values of Mean and volume had strong positive correlations between C2 and C3 (r > 0.837, $p < 0.001$) (Figure 3a,b).

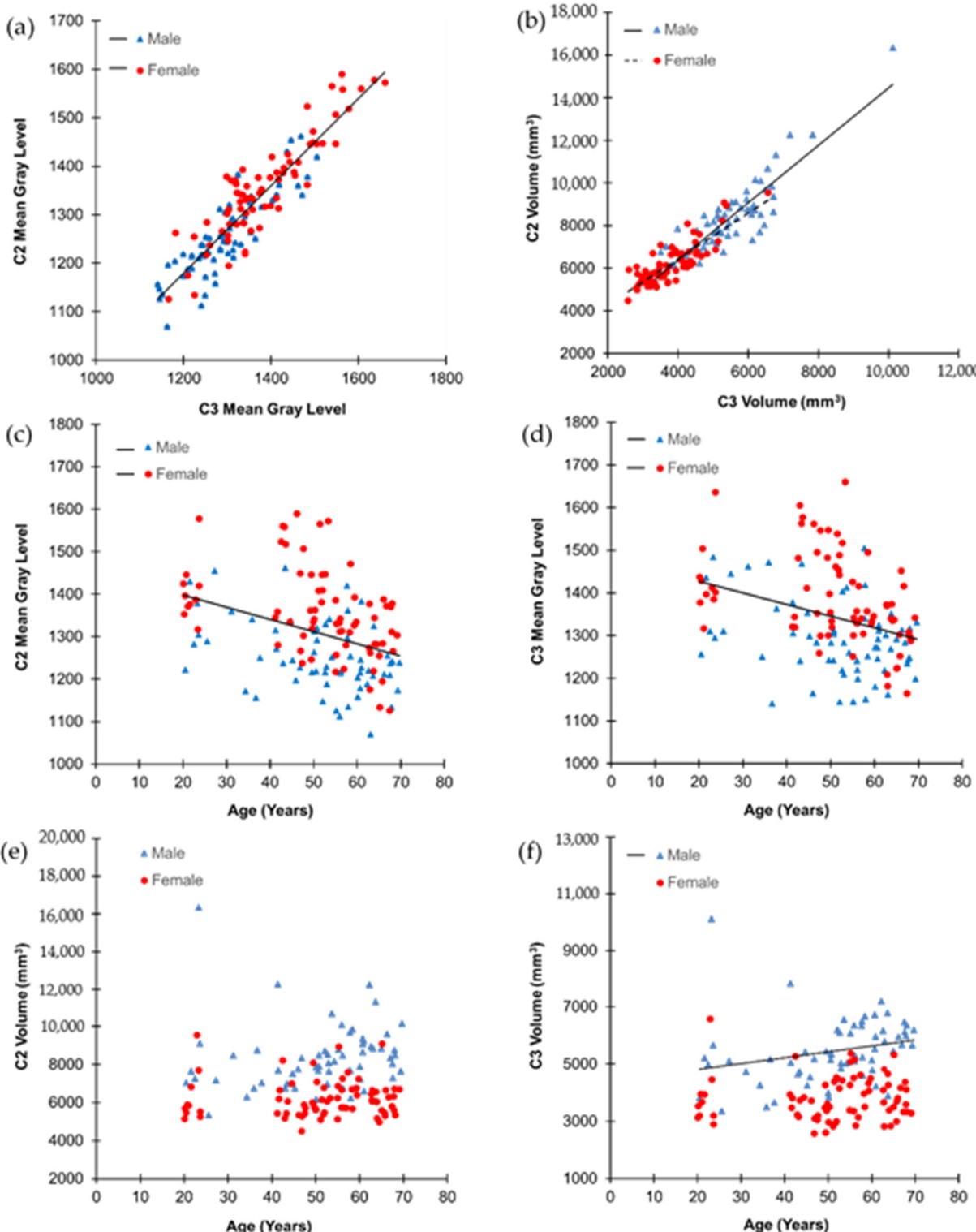

**Figure 3.** Correlations between C2 and C3 for (**a**) Mean gray level and (**b**) volume, and those of age with (**c**) C2 and (**d**) C3 Mean gray levels, and (**e**) C2 and (**f**) C3 volumes. The trend lines indicate a significant correlation ($p < 0.05$). A significant interaction of sex on the correlation of volume between C2 and C3 ($p = 0.033$) but not all other significant correlations ($p > 0.118$).

The values of Mean and volume had significant positive correlations with most of other gray level and morphological parameters of C2 and C3 in both male and female, respectively ($p < 0.01$).

Most of the gray level parameters had significantly higher values of C2 and C3 in female than in male for all age groups ($p < 0.039$) except Mean and Low$_5$ values of C3 for 60-age group between male and female ($p > 0.055$) (Figures 2g, 3 and 4). On the other hand, most of the morphological parameters had significantly higher values of C2 and C3 in male than in female for all age groups ($p < 0.035$) except concavity of C2 for all age groups and C3 for 40- and 50-age groups between male and female ($p > 0.131$) (Figure 4).

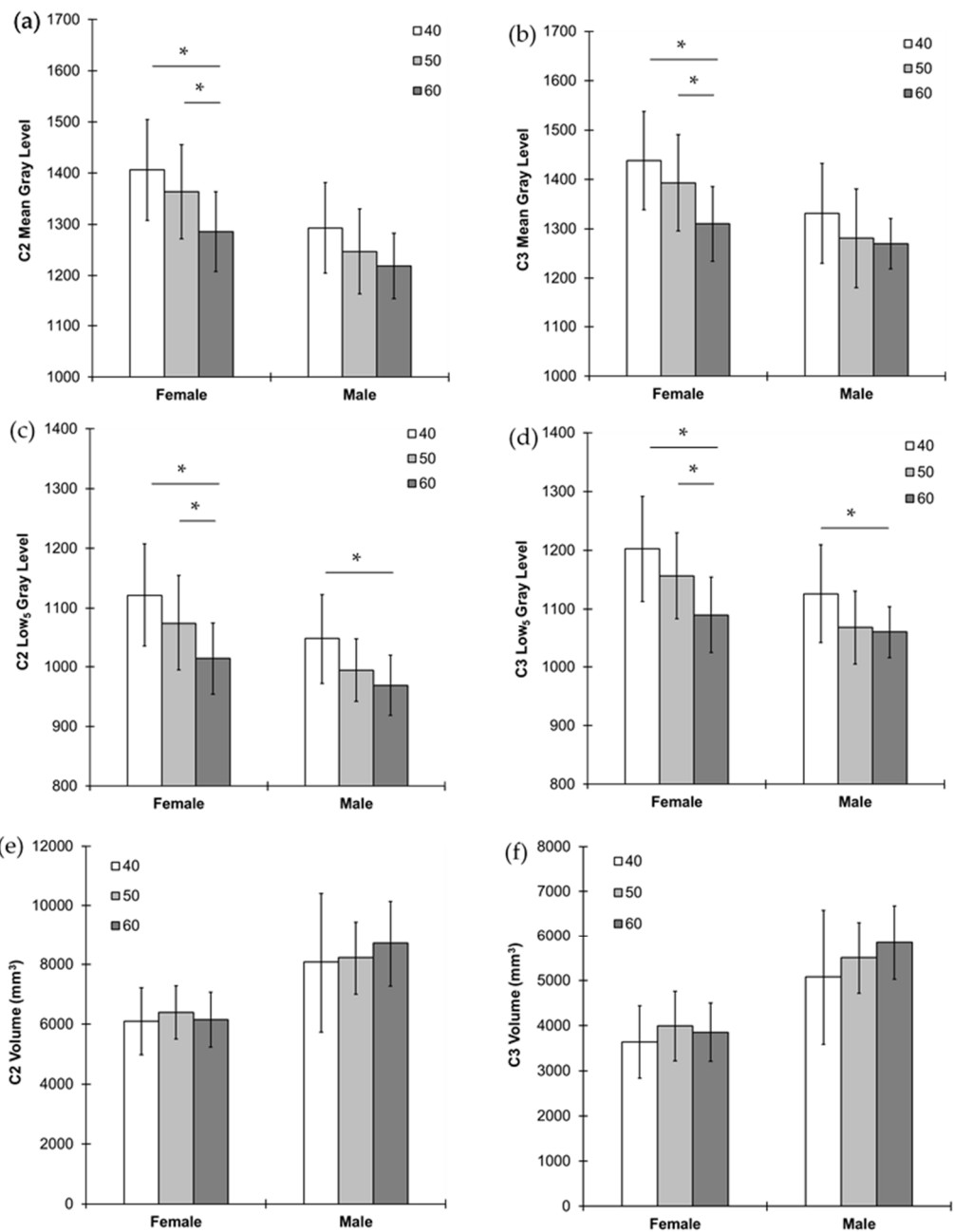

**Figure 4.** Comparisons between age groups of younger than 50 (40), 50s (50), and older than 50 (60) years in each female and male for (**a**) C2 and (**b**) C3 Mean gray levels, (**c**) C2 and (**d**) C3 Low5 gray levels, and (**e**) C2 and (**f**) C3 volumes. * $p < 0.05$.

Age had significant negative correlations with the values of Mean, Low$_5$, and High$_5$ of C2 and C3 in both male and female ($p < 0.05$) (Figure 4c,d). Males had significant positive correlations of age with volume and width of C2 and C3 ($p < 0.05$) while females had a negative correlation with height of C2 ($p = 0.04$). The female 60-age group had significantly

lower values of Mean and $Low_5$ of C2 and C3 than both the female 40- and 50-age groups ($p < 0.03$) (Figure 4). The values of $High_5$ of C2 and C3 were significantly lower for the female 60-age group than the female 40-age group and both female 40- and 50-age groups, respectively ($p < 0.015$). The values of these parameters were not significantly different between the female 40- and 50-age groups ($p > 0.34$). The male 60-age group had significantly lower values of $Low_5$ and width of C2 and C3 than the male 40-age groups ($p < 0.05$) (Figure 4). All other comparisons and correlations were not significant ($p > 0.05$).

## 5. Discussion

The current study found that the female cervical vertebrae (C2 and C3) had higher bone mineral density (BMD) but smaller size than the male C2 and C3, respectively. The cervical vertebral BMD substantially decreases in the female 60-age group, that is likely in postmenopause, while their size was maintained the same between age groups. On the other hand, the male age groups showed no changes of BMD and size between age groups. These results were consistent with previous clinical studies using QCT for human cervical and lumbar vertebrae [7,8]. As such, it is suggested that the current dental CBCT based analyses of the cervical vertebral BMD distribution and size can provide useful information to diagnose osteoporosis.

### 5.1. Evaluation of Methodology

It has been a long-standing debate about whether the CBCT can assess BMD. The main issue stemmed from the reliability and consistency to assess the X-ray attenuation using Hounsfield unit (HU) [25–27]. However, the current CBCT machine is successfully calibrated with the strong positive correlations between hydroxyapatite phantoms and gray levels (Figure 1) [11]. The current study used the gray levels without converting them to BMD because the absolute BMD values are different between CBCT machines depending on their calibration tools [17]. If the gray levels are converted using the equation in Figure 1, the Mean gray level values of C3 female 40- and 60-age groups are corresponding to BMD values of $1600.41 \pm 115.7$ mg/cm$^3$ and $1458.36 \pm 86.14$ mg/cm$^3$, respectively. Those values are at comparable range of integral volumetric BMD for human lumbar vertebrae measured by 3D QCT [7].

As BMD has significant correlations with mechanical properties of vertebrae, it has been considered as an important surrogate to estimate the risk of bone fracture due to osteoporosis [2,7–9,28]. In addition to BMD, the current study investigated more parameters including its distribution and morphology of the cervical vertebrae. The heterogeneous distribution of BMD in the vertebra is resulted from bone remodeling that occurs at different time points. In particular, estrogen deficiency at menopause triggers active bone resorption followed by formation [3]. As bone resorption outbalances bone formation, a net bone loss occurs in postmenopausal osteoporosis [2,3]. As a result, more mineralized pre-existing bone tissues are resorbed, and less mineralized newly formed bone tissues are formed producing reduction of overall BMD. The gray level histogram in the current study reflects on this biological change of minerals in bone (Figure 2). Resorption of the pre-existing bone tissues decreases the value of $High_5$ while formation of new bone tissues decreases the value of $Low_5$ [23]. The value of Mean gray level is determined in association with those of $High_5$ and $Low_5$ as accounted for their strong positive correlations.

### 5.2. Clinical Application

Consistent with the current findings, volumetric BMDs of human C2 and C3 showed a strong positive correlation in the previous study using QCT [9]. Furthermore, many QCT based human studies have observed that females have higher BMD and smaller size of cervical and lumbar vertebrae than males [7,8]. These observations support reliability of the current dental CBCT based BMD and morphological analysis. The bigger cross sectional area and volume of male lumber vertebra has an advantage to bear more axial static loading due to body weight [7,29]. In contrast, it is indicated that incidence of fall-induced cervical

vertebral injury is more in males than females [6,30]. These results provide an insight that higher BMD in female cervical vertebra may provide more resistance to fall-induced impact loading than bigger size of male cervical vertebra. Further studies are needed to clarify the relationships between functional demands and characteristics at different anatomical sites of spine.

While both males and females had the similar reducing trend of gray level parameters with advancing age, females showed significant rapid reduction between 50- and 60-age groups, which was not observed in males. These findings suggest that the altered gray level parameters of female 60-age group likely results from rapid decrease in BMD due to menopause while males experience progressive decrease in BMD due to senile osteoporosis as observed in other anatomical sites [1].

The gray level parameters and volume of cervical vertebrae were computed based on 3D images. The volume had significant correlations with other morphological parameters that changed slightly with ages. These findings indicate that the structure of cervical vertebrae was maintained with ages in adult patients examined in the current study. In addition, the SD of gray level, which represents heterogeneity of BMD and changes by adding or losing bone tissues, was also consistent with age, indicating that the absolute values of BMD decreased without bone loss.

A limitation of the current study is that cross-sectional analysis was performed using the existing patients' CBCT images. A longitudinal study for individual patients at different time periods of age is expected to provide more significant results for progressively developing osteoporosis with aging. However, multiple radiographic scanning for the same patient is limited due to potential accumulation of radiation dose. Another limitation may arise from that the current CBCT images were investigated retrospectively. The current findings would be more informative if they are compared with those from other anatomical sites using the standard BMD measurement with DEXA and the same methodologies used in the current study. However, it is not available to rescan the same patients. Instead, we found a previous study showing that a radiographic density obtained from a 2D slide of CBCT image for the cervical vertebrae had significant correlations with DEXA based osteoporotic scores at lumbar and femoral neck [31]. Moreover, 3D QCT based BMD of C2 and C3 showed significant correlations with those of thoracic and lumbar vertebrae [9]. These results suggest further studies that can be designed to compare BMD and morphology of cervical vertebrae using dental CBCT and those of other vertebrae using medical CT.

In conclusion, this is the first study that examined capability of dental CBCT to assess the integral volume BMD of human cervical vertebrae. The current study provided evidence that CBCT based analysis may suggest additional information to detect the rapid BMD reduction of C2 and C3 at the postmenopausal age. As the CBCT is widely used for routine dentistry including dental implant planning, visualization of abnormal teeth, evaluation of the jaws and face, cleft palate assessment, diagnosis of dental caries, endodontic diagnosis, and diagnosis of dental trauma as indicated by FDA (Food & Drug Administration) [32], it is an easily accessible tool to provide additional health information to patients who have taken CBCT for other dental treatments. As such, the strength of the current study is to produce a baseline data that can be used to develop CBCT technique to be a future diagnostic tool of osteoporosis. The knowledge of cervical vertebral BMD and morphology is also helpful for surgeons to obtain a better treatment plan of cervical vertebral degeneration and osteoporotic fracture. It was indicated that cervical spine is more vulnerable to fracture than lumbar spine [33] and elderly patients can have cervical spine damage with a minor trauma to head and neck sites [29]. Further studies need to investigate relationships of the CBCT based BMD and risk assessment of cervical spine with existing knowledge based on lumbar spine.

**Author Contributions:** Conceptualization, D.-G.K. and E.-S.M.; methodology, D.-G.K., E.-S.M., S.K., N.K. and M.J.; software, D.-G.K., E.-S.M., S.K., N.K. and M.J.; validation, D.-G.K. and E.-S.M.; formal analysis, D.-G.K. and E.-S.M.; investigation, D.-G.K. and E.-S.M.; resources, D.-G.K., T.D., F.Z. and D.J.L.; data curation, D.-G.K. and E.-S.M.; writing—original draft preparation, D.-G.K. and E.-S.M.; writing—review and editing, D.-G.K., T.D., F.Z. and D.J.L.; visualization, E.-S.M., S.K., N.K. and M.J.; supervision, D.-G.K.; project administration, D.-G.K.; funding acquisition, D.-G.K. and E.-S.M. All authors have read and agreed to the published version of the manuscript.

**Funding:** This study was, in part, supported by student research program, College of Dentistry, The Ohio State University.

**Institutional Review Board Statement:** The protocol used in the current study was approved by institutional review board at Ohio State University (Protocol no. 2011H0128).

**Informed Consent Statement:** Informed consent was obtained from all subjects involved in the study.

**Data Availability Statement:** Not applicable.

**Acknowledgments:** We thank Jie Liu and Keiichiro Watanabe for helping with data collection.

**Conflicts of Interest:** The authors declare no conflict of interest.

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
