# Peer review of "Aging Alters Cervical Vertebral Bone Density Distribution: A Cross-Sectional Study"

_applsci, doi:10.3390/app12063143_

Round 1

Reviewer 1 Report

The presented paper shows a highly interesting field of possible further use of dental CBCT. Especially considering possible second use of already acquired data, originally gathered for other purposes this presents a promising idea of osteoporosis screening. Especially if a specific software doing this as a sort of a background check on already acquired medical images is designed and applied.

However, the current title of the paper is rather strong. It implies that it is first to show such an application and/or first demonstrate the correlation between gray level parameters and aging. I understand that the goal probably was to have a descriptive title, but this one is really too strong. I have also found the original author's thesis (https://etd.ohiolink.edu/apexprod/rws_etd/send_file/send?accession=osu1616595575448557&disposition=inline) which I honestly think is very well written and I like its quality very much. I even think the original title of the manuscript proposed in the thesis is more appropriate.

As for the theoretical background, I think that the papers cited in the paper are quite old, even though, in this specific case, a lot of newer publications have been made. One of the newer studies is described here (10.30485/ijsrdms.2019.89758). The most recent one is for example here (10.1016/j.oooo.2021.06.014). Unfortunately, none of them are cited by the authors, nor do they comment on those results.

Also, some of the results have been validated on CT as well (http://dx.doi.org/10.11606/issn.2357-8041.clrd.2019.155263). Maybe the authors will be able to also add their own results from the mandibula (posterior region) or anterior maxilla to compare. Possibly adding citations of the aforementioned papers and commenting and comparing them with your own results will make the paper more up-to-date.

If I should comment on the methods used by the authors, currently, even fractal methods (also in the context of postmenopausal women) were tested in that matter, though they were only partially successful (10.5624/isd.20210172). Nevertheless, I have nothing to say against the methods or the quality of the text itself. Just the novelty of the theoretical background could be improved.

Reviewer 2 Report

Some issues are present in the manuscript:

- lack of information about the correlation between Gray level and BMC: only in the discussion a paper is cited (ref. n. 11) that address this aspect. To me is the starting point, and it can be at least cited as a reference from other studies. The author can add this part at the beginning of the manuscript.

- in general there is confusion between the innovative method and clinical results. It is not clear to me what the scope of the publication is : the new methodology or the clinical results .

I suggest organizing the manuscript and the discussion splitting the two aspects more logicallly: at the moment for me the article is not clear.

- many statistical tests and results are reported : statistical analysis are of different types and the reader sometimes has difficulty understanding which type of analysis is actually performed.

Data can be organized differently to help the reader to understand the results .

Personally I prefer Fig.2 or Fig.3 to Table 1 or 2 which are unreadable to me.

- I suggest more emphasis on the pro of  CBCT imaging versus DEXA (a clear comparison is not present in the article and I assume DEXA is currently the gold standard for BMC evaluation) . Something is present in the discussion but can be moved to the beginning.

You can add a simple table summarizing the main differences : 2D/3D, spatial resolution, FOV.

-No information on radiation dose : in the introduction high dose is attributed to CT (correct), and the low dose to DEXA . In the previous table the authors can add this point al well.

Reviewer 3 Report

I would like to congratulate the authors for conducting the present study.

The study design, it seems a cross-sectional study, should be mentioned in the Title.

I recommend placing the keywords in alphabetic order.

The introduction looks fine and the aim sentence acceptable.

May the authors describe the patients recruitment method? Were all included? Or a part of the overall subpopulation?

What was the sampling method? Is this a convenience sample?

May the authors provide more demographic data such as average age or most relevant ethnic group?

When were the CBCTs performed? Was any CBCT performed with the solo proposed of this study?

When were the CBCT volumes screened?

Where there any CBCT volumes excluded during the assessment? What was the exclusion rate?

Were there any possible source of  bias detected? And how were they managed?

I suggest the authors do address the study strength?

I recommend the authors to address the study generalization and external validity.

I recommend the authors to review the references, some are not according the journal guidelines.

Round 2

Reviewer 3 Report

Dear authors, i have no more concerns. Thank you.